# High Prevalence of Persistent Measurable Postoperative Knee Joint Laxity in Patients with Tibial Plateau Fractures Treated by Open Reduction and Internal Fixation (ORIF)

**DOI:** 10.3390/jcm12175580

**Published:** 2023-08-27

**Authors:** Markus Bormann, Claas Neidlein, Niels Neidlein, Dennis Ehrl, Maximilian Jörgens, Daniel P. Berthold, Wolfgang Böcker, Boris Michael Holzapfel, Julian Fürmetz

**Affiliations:** 1Department of Orthopaedics and Trauma Surgery, Musculoskeletal University Center Munich (MUM), University Hospital, LMU Munich, Marchioninistraße 15, 81377 Munich, Germany; 2Division of Gastroenterology, Brigham and Women’s Hospital, Harvard Medical School, Boston, MA 02115, USA; 3Department of Hand, Plastic and Aesthetic Surgery, LMU Munich, 81377 Munich, Germany; 4Department of Trauma Surgery, Trauma Center Murnau, 82418 Murnau am Staffelsee, Germany

**Keywords:** tibial plateau fractures, instability, post-traumatic osteoarthritis, tibial slope

## Abstract

The development of post-traumatic osteoarthrosis after tibial plateau fracture (TPF) is multifactorial and can only be partially influenced by surgical treatment. There is no standardized method for assessing pre- and postoperative knee joint laxity. Data on the incidence of postoperative laxity after TPF are limited. The purpose of this study was to quantify postoperative laxity of the knee joint after TPF. Fifty-four patients (mean age 51 ± 11.9 years) were included in this study. There was a significant increase in anterior–posterior translation in 78.0% and internal rotation in 78.9% in the injured knee when compared to the healthy knee. Simple fractures showed no significant difference in laxity compared to complex fractures. When preoperative ligament damage and/or meniscal lesions were present and surgically treated by refixation and/or bracing, patients showed higher instability when compared to patients without preoperative ligament and/or meniscal damage. Patients with surgically treated TPF demonstrate measurable knee joint laxity at a minimum of 1 year postoperatively. Fracture types have no influence on postoperative laxity. This emphasizes the importance of recognizing TPF as a multifaceted injury involving both complex fractures and damage to multiple ligaments and soft tissue structures, which may require further surgical intervention after osteosynthesis.

## 1. Introduction

The incidence of tibial plateau fractures (TPFs) has increased by up to 68% in the last decade [1,2,3,4,5]. In recent years, computed tomography scans (CT) became the gold standard in the primary diagnostic of TPF [6,7]. Thus, the more comprehensive depiction of the fracture led to the development of new classification systems and altered the surgical approaches for tibial plateau fractures [8,9]. The choice of surgical approach for osteosynthesis is mostly made depending on the type of fracture as well as concomitant ligamentous and (osteo-)chondral or meniscal injuries, and treatment involving anterior, posterior, and arthroscopic approaches may be required [8].

Although there are various surgical options to choose from, 23 to 44% of the patients develop post-traumatic osteoarthrosis (PTA), leading to 3 to 7% of the patients requiring total knee arthroplasty (TKA) within 10 years after sustaining TPF [10,11,12,13,14,15,16,17]. Interestingly, some authors report worse functional outcomes in patients with TKA after sustaining PTA when compared to patients with TKA indicated for primary osteoarthritis (OA) [18,19,20]. However, the development of PTA is multifactorial and has not yet been definitively clarified [21,22,23]. These factors include cartilage lesions, meniscal lesions, post-traumatic knee joint laxity, post-traumatic joint step, post-traumatic axial deviation, and/or widening of the tibial plateau, as well as higher age [21]. Osteosynthetic treatment aims to anatomically reconstruct the joint surface, leg axis, and width of the tibial plateau to minimize these risk factors [24,25]. However, meniscal and ligamentous injury factors often go unrecognized in tibial plateau fractures [26], as to date there is no strong recommendation for using additional (magnetic resonance imaging) MRI imaging in patients following TPF [27].

In this context, concomitant meniscal and ligament injuries are recognized as individual risk factors associated with early TKA following TPF [12]. It has been widely shown that persistent knee joint laxity leads to a subjectively worse outcome [28,29,30], and that persistent knee joint laxity is a relevant risk factor for the development of PTA [21,22,23]. In addition, studies of ligamentary knee surgery show that an unstable knee joint also leads to a subjectively worse outcome [28,29,30]. Several studies show how different devices can measure laxity of the knee joint [31,32]. Consequently, this comes along with higher complication rates, while the number of (semi-)constrained prostheses after PTA is higher in this patient cohort [10,13,15,18,19].

Unfortunately, no clear recommendation has been established of how to best diagnose and treat intraoperative knee joint laxity in TPF. A German guideline recommends testing ongoing intraoperative laxity testing after osteosynthetic treatment [27]. However, it is not specified whether the examination should be conducted using fluoroscopic stress images or solely through clinical examination. The guideline suggests additional peripheral stabilization only for cases with significant medial and/or lateral laxity. Thus, the objective of this study is to investigate knee joint laxity after the surgical treatment of tibial plateau fractures using ORIF. The hypothesis was that most patients would show ongoing knee joint laxity using a comprehensive and meticulous testing apparatus.

## 2. Materials and Methods

### 2.1. Patient Selection

A retrospective chart review was performed on patient data collected at a Level I trauma center between February 2014 and March 2020. Institutional review board approval was obtained before the initiation of the study. Patients eligible for study inclusion were those aged ≥18 years treated surgically for TPF; unilateral fracture; isolated fracture of the affected leg; preoperative X-ray and/or computed tomography; and repaired or intact ligament status of the affected knee. Ligamentous status at the time of surgery was assessed pre-surgery and after ORIF before soft-tissue closure using MRI or clinical examination by the surgeon. Information was obtained from the institutional databank, intraoperative documentation, patient’s anamnesis, and/or MRI-scan. Patients were excluded if they showed extra articular fracture, fractures other than TPF, previous ligamentous injury and/or surgery, and bilateral fractures and did not have detailed intraoperative documentation. The minimum follow-up time after surgery was 12 months. Fractures were classified according to the established system of Schatzker [33]. The fractures were additionally classified according to Moore [34] if there was radiological evidence of knee dislocation. All fractures were classified by the institutional research group, which consisted of a head of department (J.F.), a consultant (M.B.), and a scientific assistant (C.N.). Disagreements between the raters’ classifications were resolved by discussion.

This study was divided into four parts. (1) First, all patients were analyzed as a group; (2) then, they were grouped according to their fracture type: simple fractures (Schatzker Type I to III) vs. complex fractures (Schatzker Type IV to VI). (3) Third, they were subdivided into one population with additional ligament repair (e.g., ACL reconstruction, in some cases with additional meniscal repair) during the primary surgery and one population where no additional ligament repair was performed. (4) In a fourth step, the entire collective was divided into a group with ≥2 mm and a group with <2 mm anterior–posterior (AP) translation.

### 2.2. Analysis of Laxity

Dynamic valgus instability (medial deviation) was measured using the Orthelligent system (OPED GmbH, Valley, Germany). The patients were asked to perform a single leg stand with a 20–30° flexion in the knee joint and to hold that position for 20 s (Figure 1). The test was performed three times, with the average value taken.

The Laxitester (ORTEMA Sport Protection, Markgröningen, Germany) was used to assess knee laxity. With a torque of 2 N (Newton), the internal and external rotation laxity of the lower leg were obtained. The device’s accuracy has been described as 5° [31]. In addition, a Lachmeter (Equipamentos Ortopedicos LTDA, Preto, Brazil) was used to measure anterior–posterior (AP) translation in the neutral position of the lower leg, as well as internal and external laxity [31,35]. AP translation was measured from the neutral position of the knee in a 30° flexion angle to the maximum anterior tibial translation.

All measurements were taken with the ankle locked in a predetermined dorsiflexion using the trapezoidal shape of the talus. As a result, the torque generated to the foot is transferred to the lower leg (Figure 2). Range of motion was measured using a Goniometer. All tests were performed on both the injured and the healthy knee, serving as its own control group.

### 2.3. Statistical Analysis

Statistical analysis was performed using a *t*-test to compare group differences for variables with normal distribution. The Kruskal–Wallis or Mann–Whitney U test was used to analyze variables with non-normal distribution. The significance level was set at *p* < 0.05. A post hoc power analysis incorporating the total sample size, proportion of patients with TPF, and concomitant instability, with an alpha value of 0.05, demonstrated a power of 89.1% for our study. For statistical analysis and graphical depiction, RStudio (version 1.4.1717 2009–2021 RStudio, PBC, 250 Northern Ave, Boston, MA 02210, USA) was used.

## 3. Results

### 3.1. Participants

At the final follow-up, 54 patients (*n* = 32 women; *n* = 22 men) with a mean age of 51 ± 11.9 years were included in this single-center study. Fracture classification and trauma mechanisms as well as other demographic data are presented in Figure 3 and Table 1.

### 3.2. Measurements

For all patients, there were significant differences (increased laxity to injured leg) in the AP translation (mean (M) = 1.02; SD ± 1.4 mm; *p* < 0.05) and AP translation in internal rotation (0.71 ± 1 mm, *p* < 0.05) between the injured leg and the healthy side. Furthermore, there were significant differences in range of motion (*p* < 0.05). In addition, patients had a significant deficit in external rotation (M = −5.1°, SD ± 10.1; *p* < 0.05). The results are presented in Table 1.

### 3.3. Complex vs. Simple Fractures

The complex fracture group consisted of 26 complex fractures with a mean age of 48.0 ± 11.9 years at the time of surgery (*n* = 3 Schatzker Type IV; *n* = 23 Type VI). The simple fracture group consisted of 28 fractures (*n* = 25 Schatzker Type II; *n* = 3 Type III) with a mean age of 44.5 ± 12.0 years. The demographic data are presented in Table 2. The fractured leg was compared. When assessed for fracture type, complex fractures show significantly (*p* < 0.05) less external rotation (−7.4° ± 11.2 vs.−3.1° ± 15.1) compared to simple fractures. Significant differences regarding the AP translation in the Laxitester measurement were not observed. There was also no significant difference between complex and simple fractures regarding the medial deviation of the knee. The results are presented in Table 2.

### 3.4. Ligament Repair vs. No Ligament Repair

When assessed for additional ligament repair, patients were classified into two groups. A lateral collateral ligament and/or anterior cruciate ligament brace was used for ligament repair. Group 1 consisted of 24 patients undergoing ORIF with additional ligament repair (*n* = 10 Schatzker Type II; *n* = 1 Type III; *n* = 13 Type VI). The mean age of this group was 48.2 ± 11.6 years. Group 2 consisted of 30 patients with no additional ligament repair (*n* = 0 Schatzker Type I; *n* = 15 Type II; *n* = 2 Type III; *n* = 3 Type IV; *n* = 10 Type VI) The mean age of this group was 46.2 ± 12.2 years. Further demographic data are presented in Table 3.

There were significant differences between the injured legs in AP translation as well as in AP translation in internal rotation (*p* < 0.05). There was a significant difference in the internal rotation angle as well as in the external rotation angle (*p* < 0.05). Significant differences in the flexion of range of motion (*p* < 0.05) were observed, while the deficit in external rotation was significantly (*p* < 0.05) higher in the ligament repair group (−7.5° ± 10.3 vs. −3.1° ± 9.6). There was no significant difference in the number of patients receiving a preoperative MRI scan, while the no ligament repair group showed significantly higher rates of preoperative CT scans.

Surgery time showed a significant difference between the groups (*p* < 0.05), with a shorter surgery time in the ligament repair group of 9.1 ± 106.9 min.

Patients with no additional ligament repair showed significantly higher rates (*p* < 0.05) in the initial treatment with a brace.

### 3.5. AP Translation ≥2 mm vs. <2 mm 

Overall, fifteen patients (27.8%) had an AP translation ≥2 mm. Out of these, ten had a simple (Schatzker Type II) and five a complex fracture (*n* = 1 Schatzker Type IV; *n* = 4 Schatzker Type VI). The group with an AP translation ≥2 mm also showed a significantly (*p* < 0.05) increased AP translation in internal rotation (1.7 ±1.2 vs. 0.44 ±0.97, *p* = 0.00013) and in external rotation (1.2 ±1.4 vs. −0.2 ±1.1, *p* = 0.0055). Preoperative MRI scans were performed more often (*p* < 0.05) in the group with an AP translation <2 mm (25.6%) than in the group with an AP translation ≥2 mm (6.7%).

## 4. Discussion

The most important finding of this study was that the majority of the patients demonstrated significant relevant knee joint laxity 1 year postoperatively after sustaining TPF when compared to their healthy knee. In other words, the data from this study show that significantly increased anterior translation and anterolateral rotation after surgically treated tibial plateau fracture occurs when using a validated and reliable testing apparatus.

Postoperative persistent knee joint laxity is often related to several factors and multifactorial. Unaddressed meniscal or ligamentous injuries as well as malreduction may evidently contribute to this instability. Other studies previously described up to 90% of concomitant injuries such as meniscal/ligamentous lesions and lesions of the posterolateral complex, some remaining neglected [26,36,37]. As 20.4% of all patients in this study had a preoperative MRI scan, only 6.7% of the patients with an AP translation of ≥2 mm had a preoperative MRI available, emphasizing the difficulty in detecting concomitant injuries in TPF. Also, the high prevalence of simple fractures in the group with an AP translation ≥2 mm may indicate that preoperative MRI imaging is required more frequently for these fractures.

When it comes to persistent anterolateral rotatory laxity (ALRL), the anterolateral complex (ALC), involving the iliotibial band with its three layers, the accompanying Kaplan fibers, and the anterolateral ligament, has been effectively shown to offer resistance against internal rotational torques, particularly at greater levels of knee flexion [38,39,40,41,42,43,44]. As the tibial insertion of these structures is located at Gerdy’s tubercule or slightly posterior to it [38,41,43,45], it is in conflict with the standardized anterolateral approach to the tibial plateau. During this procedure, the iliotibial band, which is often detached from Gerdy’s tubercule with or without a bone flake, may cause damage to the ALC [7,33,46]. In addition, the trauma mechanism leading to TPF may also lead to injuries to the ALC, thus leading to persistent ALRL.

In this study, the anterolateral approach was the most used approach [46,47]. As such, this may be a possible explanation as to why there is no significant difference in the laxity of simple and complex fractures. In contrast, complex fractures were often treated in combination with another approach (61.4% lateral approach, 24.6% combined). This may also be an explanation for the high proportion of Schatzker Type II fractures in the cohort with an AP translation ≥2 mm, as these fractures were treated using an anterolateral approach.

In the total collective, significant instability was detected in two (AP translation and AP translation in internal rotation) of the three measured dimensions, while in the cohort with an AP translation of ≥2 mm, significant instability in all three dimensions could be observed. A measured postoperative AP translation of ≥2 mm seems to indicate that multidirectional laxity may be present.

The anatomical reduction in the joint surface, axis, and tibial slope is also an important factor for the stability of the knee joint [18,21]. During surgery, the posterolateral corner impression is frequently not visualized or dealt with properly because of inadequate visualization of the lateral imaging [48]. The resulting steeper tibial slope with an increased stress on the posterolateral corner thus contributes to persistent laxity. To avoid such complications, it is crucial to precisely reduce and stabilize the fracture, reinstate the tibial slope, and give particular attention to the posterolateral corner during the surgical procedure.

The occurrence of postoperative knee stiffness is associated with a bicondylar tibial plateau fracture [49] as well as initial treatment with an external fixator [49,50,51]. In this study, the complex fractures were initially treated significantly (*p* < 0.05) more often with an external fixator (complex fractures 34.6% vs. simple fractures 3.6%). The study’s data indicate that there is no significant difference in the range of motion (ROM) between complex and simple fractures. Nevertheless, the entire group showed a statistically significant decrease in ROM on the operated knee joint compared to the healthy side. Based on the absolute values, which show an average flexion difference of approximately 2° and extension difference of 0.6°, it is unlikely that arthrofibrosis occurred. It is still uncertain how much the postoperative adhesions in the knee joint and approach area affect not only the ROM but also the knee joint’s stability. Postoperative adhesions could also be a factor for the measured restricted external rotation of the operated knee joint compared to the healthy opposite side. This deficit is bigger in more complex fractures and fractures with ligament repair. These results conflict with other studies that have shown that external tibial rotation is increased in knees with acute cruciate ligament (ACL) and or posteromedial/posterolateral corner injuries [52,53]. Nevertheless, Mayr et al. were also able to demonstrate a reduced external rotation in their studies with ACL injured knees, though this was not significant [31,54]. It remains unclear which injuries during trauma or surgery contribute to the reduced external rotation. Biomechanical studies show the influence of pathological tibial rotation on the stability and pressure distribution of the knee joint [55,56]. The cause of the reduced external rotation should therefore be clarified in further (cadaveric) studies.

In the population of patients undergoing “ligament/meniscal repair,” the injured structures were treated with refixation/internal brace techniques. However, studies have suggested that these techniques may not achieve the same level of stability as ligament reconstruction or intact ligaments [57,58,59,60,61]. In the population “ligament/meniscal repair”, a refixation/internal brace of the injured structures was performed, and as a result, higher translation values may be expected in this population.

It is notable that surgical time for cases involving ligament repair was shorter than in the group without ligament repair. We interpret this to suggest that in situations involving complex bone injuries, the osteosynthesis procedure takes longer. Consequently, there might be a tendency to prioritize addressing the primary bone injuries in complex cases, particularly when the overall surgical time is already prolonged. This potentially implies that less attention is given to managing accompanying injuries in complex bone scenarios.

## 5. Limitations

This study comes along with some limitations inherent to its study design. First, the small sample size from this single-center study potentially creates selection bias due to exclusion criteria, in addition to the retrospective nature of the study design. Second, the study did not account for potential confounding factors. Third, the generalizability of the study may be limited due to the specific population and fracture types studied. Fourth, another important factor for the development of post-traumatic laxity is postoperative rehabilitation [20], which was not considered in this study. Fifth, several studies have shown the influence of thigh muscles on knee laxity [62,63,64,65]. However, this study did not include strength testing of thigh muscles.

Finally, undetected meniscal/ligamentous and posterolateral corner lesions may have contributed to postoperative laxity in this study.

The data from this study demonstrated a measurable, significant anterior and anterolateral laxity after surgically treated TPF, which is not adequately assessed pre- and postoperatively in terms of non-bony parameters. Although a preoperative MRI is effective for detecting such instabilities, a more accurate evaluation through clinical and instrumental examination, such as intraoperative fluoroscopic stress images and postoperative instrumental laxity tests, may be necessary. Further research is required to determine whether additional stabilization is necessary for this type of laxity, either initially or secondary during implant removal. To reduce the occurrence of PTA, it is crucial to treat TPF as a complex joint injury rather than just a fracture.

## 6. Conclusions

When treated by ORIF, patients with TPF demonstrate measurable uni- or multidimensional knee joint laxity at a minimum of 1 year postoperatively. Interestingly, fracture types (according to Schatzker and Moore) have no influence on postoperative laxity. This emphasizes the importance of recognizing TPF as a multifaceted injury involving both complex fractures and damage to multiple ligaments and soft tissue structures, which may require further intervention after osteosynthesis.

## Figures and Tables

**Figure 1 jcm-12-05580-f001:**
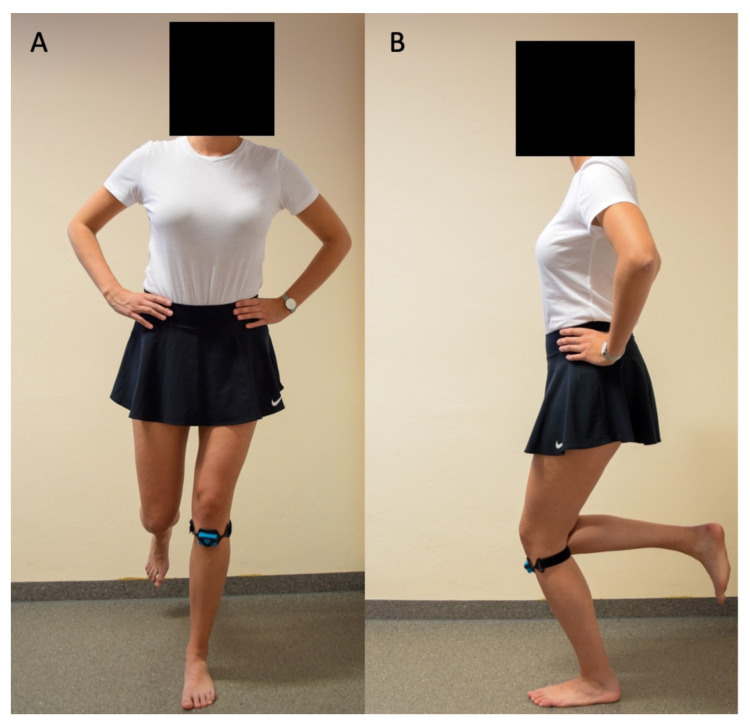
Single leg stand measuring dynamic valgus instability, (**A**) coronal view, (**B**) sagital view.

**Figure 2 jcm-12-05580-f002:**
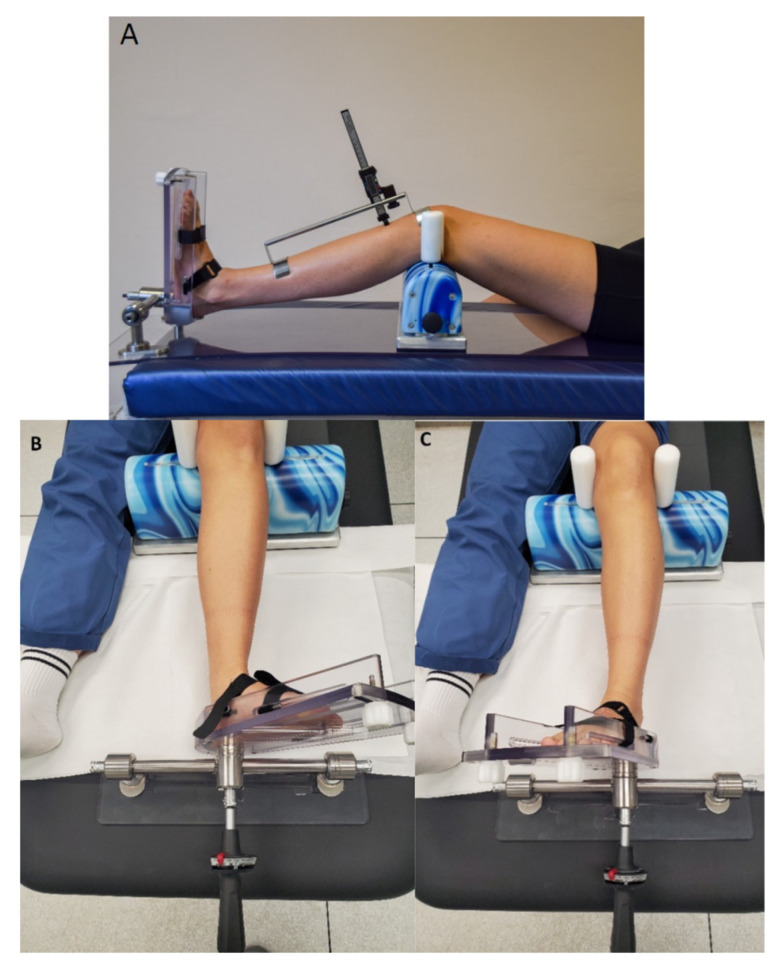
The measurement of instability was performed using Laxitester and Lachmeter (**A**) and Laxitester (**B**,**C**).

**Figure 3 jcm-12-05580-f003:**
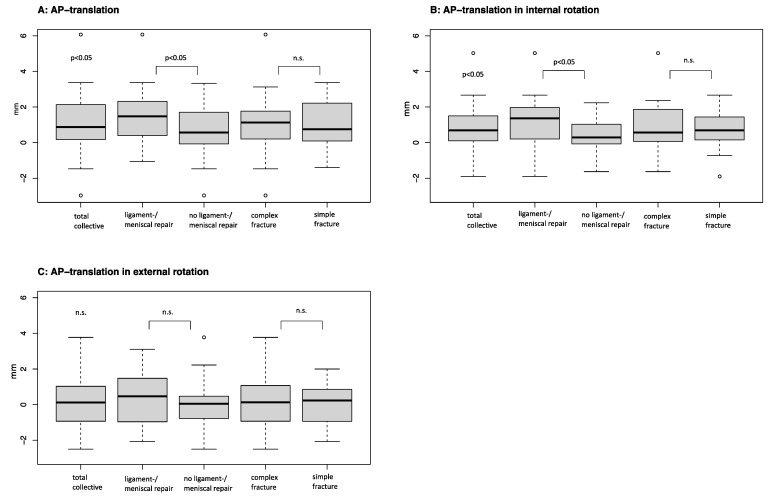
The instability of the total collective and subpopulations was depicted using boxplot for the three parameters: (**A**) anteroposterior (ap) translation, (**B**) ap translation in internal rotation, and (**C**) ap translation in external rotation. The significance level pertains to total collective = injured leg vs. healthy leg (control); subgroup ligament/meniscal repair vs. subgroup no ligament/meniscal repair; subgroup complex fracture vs. subgroup simple fracture. N.s.=not significant.

**Table 1 jcm-12-05580-t001:** Demographic data of the total patient collective. Note: results of the measurement of instability are presented as mean (95% confidence interval).

Criteria	Total Collective (*n* = 54)	*p*-Value
Male vs. female (%)	40.7% vs. 59.3%	<0.05
Mean age at surgery (years)	47 ± 11.9	-
Mean age at follow-up (years)	51 ± 11.9	-
Schatzker (*n*)		
I	0	-
II	25	-
III	3	-
IV	3	-
V	0	-
VI	23	-
Cause of accident (%)		
Falls	33.4%	-
Traffic	20.4%	-
Ski	20.4%	-
Bicycle	18.5%	-
Fall from height	0%	-
other	7.4%	-
ROM flexion (°) ^1^	127.1 ± 11.2 vs. 129.7 ± 5.3	<0.05
ROM extension (°) ^1^	2.1 ± 1.7 vs. 2.7 ± 0.8	<0.05
Initial imaging (%)		
X-ray	74.1%	-
Computed tomography	98.1%	-
Magnetic resonance imaging	20.4%	-
Initial treatment		
Brace	81.5%	-
*External fixator*	18.5%	-
Surgery time (minutes)	165.1 ± 76.8	-
ASA score	1.8 ± 0.5	-
Mean difference BMI ^2^	24.2 ± 3.1 vs. 24.9 ± 3.7	n.s.
AP translation ^3^	1.02 (0.6 to 1.5)	<0.05
AP translation in internal rotation ^3^	0.71 (0.5 to 1.1)	n.s.
Internal rotation angle ^3^	1.6° ± 10.6°	n.s.
AP translation in external rotation ^3^	0.08 (−0.1 to 0.5)	n.s.
External rotation angle	−5.1° ± 10.1	<0.05
Medial deviation ^3^	0.45	n.s.

^1^ Injured vs. healthy leg; ^2^ pre-surgery vs. follow-up; ^3^ mean side to side difference in millimeters.

**Table 2 jcm-12-05580-t002:** Population complex vs. simple fractures. Note: results of the measurement of instability are presented as mean (95% confidence interval).

Criteria	Complex Fractures (*n* = 26)	Simple Fractures (*n* = 28)	*p*-Value
Male vs. female (%)	38.5% vs. 61.5%	42.9% vs. 57.1%	n.s.
Mean age at surgery (years)	48 ± 11.9	44.5 ± 12	n.s.
Mean age at follow-up (years)	51.6 ± 11.9	50.5 ± 12	
Schatzker (*n*)			
I	0	0	-
II	0	25	-
III	0	3	-
IV	3	0	-
V	0	0	-
VI	23	0	-
Cause of accident (%)			
Falls	23.1%	42.9%	<0.05
Traffic	26.9%	14.3%	<0.05
Ski	19.2%	21.4%	n.s.
Bicycle	26.9%	10.7%	<0.05
Fall from height	0%	0%	n.s.
Other	3.8%	10.7%	n.s.
ROM flexion (°) ^1^	126.3 ± 14.1 vs. 129.8 ± 7.8	127.8 ± 7.8 vs. 129.6 ± 5	n.s.
ROM extension (°) ^1^	2.3 ± 1.8 vs. 2.9 ± 0.7	1.8 ± 1.6 vs. 2.6 ± 0.9	n.s.
Initial imaging (%)			
X-ray	69.2%	78.6%	<0.05
Computed tomography	100%	96.4%	n.s.
Magnetic resonance imaging	19.2%	21.4%	n.s.
Initial treatment			
Brace	65.4%	96.4%	<0.01
External fixator	34.6%	3.6%	<0.01
Surgery time (minutes)	173 ± 91	157.8 ± 62.1	n.s.
ASA score	2 ± 0.5	1.6 ± 0.5	-
Mean difference BMI ^2^	24.8 ± 2.6 vs. 24.8 ± 4.4	23.6 ± 3.5 vs. 24.7 ± 3.8	n.s.
AP translation ^3^	1.12 (0.4 to 1.8)	1.02 (0.5 to 1.5)	n.s.
AP translation in internal rotation ^3^	0.83 (0.3 to 1.4)	0.71 (0.3 to 1.1)	n.s.
Internal rotation angle ^3^	1.2° ± 9.6	2.9° ± 11.6	n.s.
AP translation in external rotation ^3^	0.35 (−0.2 to 1)	0.08 (−0.3 to 0.5)	n.s.
External rotation angle	−7.4° ± 11.2	−3.1° ± 15.1	<0.05
Medial deviation ^3^	0.67	0.25	n.s.

^1^ Injured vs. healthy leg; ^2^ pre-surgery vs. follow-up; ^3^ mean side to side difference in millimeters.

**Table 3 jcm-12-05580-t003:** Population: ligament repair vs. no ligament repair. Note: results of the measurement of instability are presented as mean (95% confidence interval).

Criteria	Ligament Repair (*n* = 24)	No Ligament Repair (*n* = 30)	*p*-Value
Male vs. female (%)	41.7% vs. 58.3%	40% vs. 60%	n.s.
Mean age at surgery (years)	48.2 ± 11.6	46.2 ± 12.2	n.s.
Mean age at follow-up (years)	52.5 ± 11.6	49.8 ± 12.2	
Schatzker (*n*)			
I	0	0	-
II	10	15	-
III	1	2	-
IV	0	3	-
V	0	0	-
VI	13	10	-
Cause of accident (%)			
Falls	33.3%	33.3%	n.s.
Traffic	20.8%	20%	n.s.
Ski	20.8%	20%	n.s.
Bicycle	20.8%	16.7%	n.s.
Fall from height	0%	0%	n.s.
other	4.2%	10%	n.s.
ROM flexion (°) ^1^	126.2 ± 14 vs. 129.5 ± 6.9	127.8 ± 8.4 vs. 129.8 ± 4.2	<0.05
ROM extension (°) ^1^	2.3 ± 1.7 vs. 2.7 ± 0.6	1.9 ± 1.7 vs. 2.7 ± 1	n.s.
Initial imaging (%)			
X-ray	79.2%	70%	n.s.
Computed tomography	100%	96.7%	<0.05
Magnetic resonance imaging	20.8%	20%	n.s.
Initial treatment			
Brace	75%	86.7%	<0.05
External fixator	25%	13.3%	n.s.
Surgery time (minutes)	160 ± 72.7	169.1 ± 80.7	<0.05
ASA score	1.8 ± 0.4	1.8 ± 0.6	-
Mean difference BMI ^2^	24.4 ± 3.7 vs. 24.2 ± 4.1	24 ± 2.7 vs. 25.5 ± 3.2	n.s.
AP translation ^3^	1.51 (0.9 to 2.1)	0.71 (0.1 to 1.2)	<0.05
AP translation in internal rotation ^3^	1.19 (0.6 to 1.8)	0.44 (0.1 to 0.8)	<0.05
Internal rotation angle ^3^	−0.3° ± 12.4	3.8° ± 8.5	<0.05
AP translation in external rotation ^3^	0.38 (−0.2 to 1)	0.07 (−0.3 to 0.6)	n.s.
External rotation angle	−7.5° ± 10.3	−3.1° ± 9.6	<0.05
Medial deviation ^3^	0.25	0.61	n.s.

^1^ Injured vs. healthy leg; ^2^ pre-surgery vs. follow-up; ^3^ mean side to side difference in millimeters.

## Data Availability

Not applicable.

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
