# Peer review of "High Prevalence of Persistent Measurable Postoperative Knee Joint Laxity in Patients with Tibial Plateau Fractures Treated by Open Reduction and Internal Fixation (ORIF)"

_jcm, 2023, doi:10.3390/jcm12175580_

Round 1
Reviewer 1 Report
This is an interesting article, and on a topic that is being increasingly understood regarding the relevant of non-bony injuries around plateau fractures.
Below are specific comments, however my main concern is the presentation of the results. The sample size is small (no direct criticism here, it’s a good series considering the injury type), especially when comparing sub-groups, yet only means and SD are shown. 95% confidence intervals should be being displayed instead as these would factor in the sample size. Currently, some of the conclusions seem strong based on results with relatively large overlapping SDs, and to me, having the sample size factored into the results makes them stronger regardless of what they show.

There are some general errors in structure that need to be corrected, such as limitations coming after the conclusions. There are also numerous grammatical / syntax errors within the text that need addressing.
Reviewer 2 Report
This empathizes of recognizing TPF as a multifaceted injury involving both complex fractures and damage to multiple ligaments, may requiring further ligamentous surgical intervention in the postoperative course. Some revised needs to done by authors as follows:
1. In the introduction section, the basic concept of tibial plateau fractures needs to more detail.
2. Several surgical option other than Open reduction and Internal Fixation should be provided.
3. What is the current article novel? It has been extensively discussed in the past. Nothing truly novel in its current state. The absence of anything original makes the current study seem like a replication or a modified study. The introduction section should contain specifics about the writers' uniqueness. It is a significant reason to reject this study.
4. As the methods authors using Fifty-four patients (mean age 51 ± 11.9 years), I feel it is not appropriate with relatively small number of participant and heterogeneous data. Would be possible to improved the quality of patient used.
5. The mechanism of fracture would be provided in the discussion to make it more comprehensive.
6. Please explain potential further study performing computational simulation. It brings several advantage’s such as lower cost and faster results compared to clinical and experimental. For this purpose. Provide the information along with relevant reference as follows: https://doi.org/10.3390/biomedicines11030951, https://doi.org/10.3390/ma16093298, and https://doi.org/10.1016/j.heliyon.2022.e12050
-
Round 2
Reviewer 1 Report
Good corrections.
last thing is that you don't need to show the SD if you are showing the CI, so anytime the CI is shown, remove the SD.
Author Response
Dear Reviewer 1,
we are very grateful for your helpful comments on our manuscript .We took your comment very seriously and worked on every aspect you mentioned. All changes to the manuscript are currently marked using the track changes modes. Please find our reply to your comments below, referring to the sections in the manuscript where necessary. Hereby we submit you a revised version of our manuscript.
Reviewer 1:
last thing is that you don't need to show the SD if you are showing the CI, so anytime the CI is shown, remove the SD. -done.